# Fetotoxicity of Nanoparticles: Causes and Mechanisms

**DOI:** 10.3390/nano11030791

**Published:** 2021-03-19

**Authors:** Chuanfeng Teng, Cuijuan Jiang, Sulian Gao, Xiaojing Liu, Shumei Zhai

**Affiliations:** 1School of Chemistry and Chemical Engineering, Shandong University, Jinan 250100, China; chuanfeng2100@163.com; 2School of Environmental Science and Engineering, Shandong University, Qingdao 266237, China; cjjiang@sdu.edu.cn; 3Jinan Eco-Environmental Monitoring Center of Shandong Province, Jinan 250101, China; jnhbjgaosulian@jn.shandong.cn; 4Department of Occupational and Environmental Health, School of Public Health, Cheeloo College of Medicine, Shandong University, Jinan 250012, China; liuxiaojing920226@163.com

**Keywords:** biological mechanisms, fetotoxicity, nanoparticle, physicochemical properties, transplacental transfer

## Abstract

The application of nanoparticles in consumer products and nanomedicines has increased dramatically in the last decade. Concerns for the nano-safety of susceptible populations are growing. Due to the small size, nanoparticles have the potential to cross the placental barrier and cause toxicity in the fetus. This review aims to identify factors associated with nanoparticle-induced fetotoxicity and the mechanisms involved, providing a better understanding of nanotoxicity at the maternal–fetal interface. The contribution of the physicochemical properties of nanoparticles (NPs), maternal physiological, and pathological conditions to the fetotoxicity is highlighted. The underlying molecular mechanisms, including oxidative stress, DNA damage, apoptosis, and autophagy are summarized. Finally, perspectives and challenges related to nanoparticle-induced fetotoxicity are also discussed.

## 1. Introduction

The rapid development of nanotechnology has led to the increased production and use of nanomaterials in industry, biomedicine, and everyday life [1,2,3], thus highlighting the need for a rise in biosafety evaluation, especially for susceptible populations [4,5,6,7]. Among these, pregnant women and developing embryos/fetuses are more vulnerable to xenobiotics. Embryo–fetal development is a critical window of exposure-related susceptibility because the etiology of diseases in adulthood may have a fetal origin. Due to their unique physicochemical properties, nanoparticles (NPs) have the tendency to cross the blood–placental barrier and reach the fetus, leading to various fetal abnormalities. In utero exposures to nanomaterials not only cause adverse pregnancy outcomes and intrauterine fetal development but also lead to the occurrence of adult chronic diseases [8,9].

So far, numerous studies have indisputably demonstrated that maternal exposure to nanomaterials during gestation result in fetotoxicity, including adverse prenatal effects on the fetus [10,11,12,13,14,15,16,17], neurotoxicity [18,19,20,21,22,23], reproductive toxicity [24,25,26,27,28], immunotoxicity [29,30,31,32] and respiratory toxicity [29,33,34,35] in offspring or even in adulthood [16,17,18,36]. The direct translocation of NPs or indirect interference from maternal damage-induced maternal mediators and placental mediators contributes to the fetotoxicity induced by prenatal exposure to NPs [37,38].

Here, we summarized the progress of NP-induced fetotoxicity based on the categories of NPs to identify crucial factors associated with NP-induced fetotoxicity and the mechanisms involved. We discussed how the physicochemical properties of NPs, administration schedule, gestation stages, and maternal pathophysiological conditions affected NPs’ fetotoxicity, as well as the possible underlying mechanisms, which will be a cornerstone of protecting susceptible populations from NP exposure.

## 2. NPs-Induced Fetotoxicity

In this section, NP-induced fetotoxicity, including detrimental prenatal outcomes, neurodevelopmental toxicity, reproductive toxicity, immune toxicity, and respiratory toxicity, was summarized.

### 2.1. Adverse Prenatal Effects on Gestational Parameters

Recent epidemiological studies have indicated a significant association between maternal occupational exposure to NPs and being small for gestational age [39,40]. Maternal NP exposure induces detrimental prenatal outcomes such as low birth weight, preterm birth, miscarriage, fetal resorption, morphological malformations, intrauterine growth retardation, and other embryo–fetal developmental abnormalities (Table 1).

*Ambient fine or ultrafine particles.* Air pollution is a global threat to health. The potential detrimental effects of ambient (ultra)fine particles on fetal and child development have been evidenced. For example, pregnant C57BL/6J *p*^un^/*p*^un^ female mice were intratracheally instilled with collected ultrafine particles at the accumulated dose of 400 μg/kg every other day during gestational day 7 (GD7)-GD17 to evaluate the fetal development under the adverse intrauterine environment induced by particulate matter (PM) [12]. This in utero ultrafine particle exposure induced placental dysfunction, which resulted in embryo reabsorption and a significant decrease in fetal weight. Daily nose-only exposure to diesel exhaust (DE) damaged placental function and induced early signs of growth retardation in a pregnant rabbit model [41]. However, another research suggested that it is the toxic chemicals but not the NPs in NP-rich diesel exhaust (NR-DE) or filtered diesel exhaust (F-DE) might be responsible for disrupting steroid hormone production in the corpus luteum and adrenal cortex in pregnant rats, which showed fetal weight and decreased fetal crown–rump length [42]. Recently, a study has shown that black carbon (BC) particles, a component of combustion-related particulate matter (PM), can reach the fetal side of the human placenta (Figure 1), which suggested that ambient particulates could be transported towards the fetus and represented a potential mechanism explaining the detrimental health effects of pollution from early life onwards [36].

*Carbon-based nanomaterials.* Carbon-based nanomaterials, such as carbon black (CB), carbon nanotubes (CNTs) and fullerenes are produced and widely used in the industry. Their biosafety has become a major concern. When pregnant C57BL/6BomTac mice were administered with CB NPs (14 nm) via lung exposure, both inhalation exposure (42 mg/m^3^, 1 h/day) from GD8–GD18 (11, 54 and 268 mg/mouse) and intratracheal instillation induced maternal inflammation [43]. Maternal exposure during pregnancy to CNTs, including single-walled carbon nanotubes (SWCNTs), multi-walled carbon nanotubes (MWCNTs), and functionalized CNTs, increased early miscarriages, fetal resorption rate, external defects, and skeletal abnormalities in fetuses. For instance, pregnant ICR mice were given a single intraperitoneal (2–5 mg/kg) or intratracheal (3–5 mg/kg) administration of MWCNTs on GD9 to assess embryo–fetal developmental toxicity. The results showed that both exposure routes caused various types of morphological and skeletal malformations. Short or absent tails and fusion of the ribs or vertebral bodies were observed in all intraperitoneally treated groups; these malformations were also observed in intratracheal exposure groups with a higher dose (4/5 mg/kg) [44]. Both intratracheal instillation and intravenous injection of MWCNTs (100 μg/kg) from GD17–GD19 led to decreased fetal weight [45]. Qi et al. reported that oxidized MWCNTs, which primarily accumulated in the liver, lung, and heart of the fetus, caused fetal development delay, fetal heart and brain disruption, and fetal resorption. Furthermore, maternal exposure to MWCNTs might induce transplacental mutagenesis [70]. Other reports showed that the intratracheal instillation of MWCNTs at a single dose (67 μg/mouse) interfered with ovulation; however, no significant alterations in litter or gestational parameters, such as implantations, sex ratio, litter size, and implantation loss, were observed [26,71]. The repeated oral administration of MWCNTs (8, 40, 200, or 1000 mg/kg) during organogenesis (GD6-GD19) did not induce alterations in gestational parameters [72,73]. These results indicated that NP administration routes affected fetal toxicities, with oral administration or via inhalation showing less toxicities than intravenous injection.

The surface modification of NPs could affect nanotoxicity. When pregnant mice were intravenously injected with amino-functionalized polyethylene glycol (PEG)-SWCNTs at a single dose of 0.1, 10, and 30 μg/mouse on GD6 or in multiple administrations of 10 μg/mouse on GD6, GD9, and GD12 (total dose of 30 μg/mouse), occasional teratogenic effects were observed only at a dose of 30 μg/mouse, possibly due to their transfer to the fetal–placenta unit. Functionalized SWCNTs significantly increased fetal resorptions, morphological abnormalities and skeletal deformities following a single oral exposure of 10 mg/kg on GD9. Ultraoxide-functionalization aggravated the toxicity of SWCNTs considerably with the higher induction of early miscarriages and fetal resorptions compared with pristine or oxidized SWCNTs [17]. The intravenous administration of reduced graphene oxide (rGO, 6.25, 12.5, and 25mg/kg) nanosheets into pregnant dams in the late gestational stage (~GD20) induced much more miscarriage than in the early gestational stage (~GD6) [46]. These results indicated that surface modification, administration routes, and gestational stages were the main factors contributing to the embryotoxic development of NPs.

*Metal oxide NPs.* Metal oxide NPs, especially zinc oxide NPs (ZnO NPs) and titanium dioxide NPs (TiO_2_ NPs), have become one of the most promising nanomaterials due to their wide application in sunscreens, food, biomedicine, and environmental remediation. Thus, their potential toxicity after maternal exposure has gained more attention. The oral administration of zinc oxide (ZnO) NPs, once daily by gavage during the organogenesis period, retarded fetal growth, reduced live fetal number, and increased fetal resorption and abnormalities [11,47,48,49,50,51]. Chen et al. found similar embryotoxicity at a dose of 180 or 540 mg/kg [47]. ZnO NPs induced such adverse effects in a size-dependent (13 nm and 57 nm, 200 mg/kg) and gestational stage-dependent manner (GD7–GD16, GD1–GD10) after a 10-day oral exposure [48]. When pregnant mice were intravenously injected with ZnO NPs (44 nm, 5, 10, and 20 mg/kg) from GD6–GD20, an increased number of dead fetuses, post-implantation loss, and reduced fetal weight were observed in the 20 mg/kg treatment group [49]. These results indicated that maternal exposure to ZnO NPs during pregnancy might pose health risks to both pregnant mothers and fetuses, which were dependent on the size of ZnO NPs, exposure time, exposure dose, and administration routes. Notably, the toxicity induced by zinc ion dissociated from ZnO NPs needs to be considered, in particular for oral administration.

Maternal exposure to TiO_2_ NPs during pregnancy induced alterations in gestational parameters, such as increased fetal resorption rate, smaller uteri and fetuses, and skeletal abnormalities in fetuses [16,52,53,74]. When intravenously injected into pregnant BALB/c mice, TiO_2_ NPs and SiO_2_ NPs (70 nm) significantly enhanced the fetal resorption rate, and decreased the fetal weight and uterine weight [16], which were dependent on the size, surface modification of NPs, and exposure stages of pregnancy [16,54]. Time-mated mice (ICR) were orally administered with TiO_2_ NPs (25, 50 and 100 mg/kg) during the whole gestation to investigate their adverse effects on the developing fetus [53]. TiO_2_ NPs transferred through the placenta, accumulated in the fetus and suppressed fetal development. Maternal inhalation exposure to TiO_2_ NPs (21 nm) during organogenesis decreased pup weight due to reduced placenta efficiency. However, in another study, no alterations in gestational parameters, such as litter size, fetal weight, sex distribution, and implantations, were observed [74]. The possible reason might be different sizes for adopted TiO_2_ NPs. The maternal–fetal transfer and accumulation of food-grade TiO_2_ NPs in the human placenta and meconium emphasized the need for the risk assessment of chronic exposure to TiO_2_-NPs during pregnancy [75].

Maternal exposure to other metal oxide NPs during pregnancy such as CeO_2_ NPs and CdO NPs also led to poor pregnancy outcomes [10,55]. The intravenous injection of CeO_2_ NPs into pregnant mice at a dose of 5 mg/kg daily from GD5–GD7 resulted in aberrations in decidualization, which exhibited “ripple effects” leading to fetal loss, fetal growth retardation, placental dysfunction, and even infertility [10]. When pregnant mice (ICR) were exposed to CdO NPs (11–15 nm) via nose-only inhalation either every other day (100 mg/m^3^) or daily (230 mg/m^3^) for 2.5 h/exposure from GD5–GD17, the accumulation of CdO NPs in placenta decreased the fetal length and delayed neonatal growth without direct NP transfer to the fetus [55].

*Metal NPs.* The extensive application and large-scale production of metal NPs, such as Ag NPs and Au NPs, raise safety concerns in the vulnerable stages of life. A single intravenous administration of polyvinylpyrrolidone (PVP) or citrate-coated Ag NPs (20 or 110 nm, 700 μg/kg) in the late stages of pregnancy produced moderate size- and vehicle-dependent alterations in vascular tissue contractility, suggesting possible fetal growth retardation [56]. Nose-only inhalation exposure to freshly produced Ag NP aerosol (18–20 nm, 1 or 4 h/day) during GD1–GD15 induced the translocation of Ag NPs and increased the number of resorbed fetuses [14]. The intravenous injection of 10 nm Ag NPs (66 mg/mouse) daily on GD7, GD8, and GD9 led to significant Ag accumulation in the maternal liver, spleen, and visceral yolk sac, and might potentially affect embryonic growth without accumulation in embryos/fetuses [57]. Au NPs (2, 15, and 50 nm) disturb embryonic development in a size- and concentration-dependent manner. Au NPs (15 nm) downregulated the expression of distinct germ layer markers and suppressed the differentiation of all three embryonic germ layers, consequently leading to fetal resorption [58].

In summary, in utero exposure to NPs induced potential fetal developmental toxicity or teratogenicity, such as fetal growth retardation, decreased litter size, fetal deformities, and fetal resorption. NP-induced embryo–fetal developmental toxicity depended on not only the physiochemical properties of NPs but also the exposure schedule, including administration doses and times, as well as maternal pathophysiological conditions.

### 2.2. Neurotoxicity

Maternal exposure to xenobiotics usually causes the most common developmental abnormalities in humans, especially those associated with neurodevelopment in fetus/offspring. Previous studies demonstrated that ultrafine particles/NPs penetrated through the blood–brain barrier, reached the brain, and induced neurodevelopmental toxicity [76,77,78,79]. With the incomplete blood–brain barrier, fetal brains are more vulnerable to xenobiotic pollutants, leading to pathological abnormalities in fetal brain tissues and neural damage in offspring brains, and even behavioral abnormalities in adulthood. Prenatal exposures to DEP [20,80,81], CB NPs [22,82,83,84], CNTs [70], ZnO NPs [85], TiO_2_ NPs [19,23,86,87,88,89], and Ag NPs [90] induced various organic and functional damages to the central nervous system (CNS) (Table 2).

*Ambient fine or ultrafine particles.* Epidemiological studies indicated a positive association between air pollution and autism spectrum disorder, attention-deficit hyperactivity disorder, cognitive impairment, and decreased corpus callosum (CC) volumes in children [100,101,102]. Evidence shows that (ultra)fine particle exposure during pregnancy induces neurodevelopmental abnormalities in fetuses or offspring [21,77]. Developmental exposure during the critical window of CNS development, equivalent to human first and second trimesters, produced a range of adverse neurodevelopmental disorders, including male-predominant behavioral neurotoxicity and glial activation, ventriculomegaly, depressive behaviors, impaired contextual memory, and reduced food-seeking behavior [78,79,91,103]. Prenatal air pollution exposure caused hippocampal vascular leakage and impaired neurogenesis, which was associated with behavioral deficits [78]. In utero exposure to PM_2.5_ (6 h/day, 5 days/week) altered the neuroimmune phenotype [79]. Gestational DEP exposure caused cellular and axonal hypertrophies in olfactory tissues and affected monoaminergic neurotransmission in fetal olfactory bulbs, leading to altered olfactory-based behaviors [80]. Pregnant mice were exposed 6 h/day to concentrated ambient fine/ultrafine particles at the average concentration of 92.69 μg/m^3^ during GD1–GD17 to recognize the critical window of neurodevelopmental disorders. Such exposure induced ventriculomegaly, increased the CC area, and decreased the hippocampal area in both sexes of offspring. Meanwhile, both sexes exhibited CC hypermyelination, increased microglial activation, and decreased total CC microglia. Oxidative stress and neuroinflammation are the main mechanisms of CNS toxicity induced by ambient ultrafine particles during gestation [91]. Exposure to fine and ultrafine particles during the fetal period alters postnatal oligodendrocyte maturation, proliferation capacity, and myelination [103].

*Carbon-based nanomaterials.* Intratracheal instillation, intranasal instillation, and airway exposure were often adopted to evaluate neurodevelopmental disorders induced by CB NPs. Maternal inhalation exposures to CB NPs triggered habituation pattern and brain blood vessel alteration, dysregulated gene expression in the frontal cortex, and reactive astrogliosis in fetal brains [22,82,83,84]. For example, the intranasal administration of CB NPs (2.9, 15, and 73 μg/kg) into pregnant ICR mice on GD5 and GD9 induced dose-dependent reactive astrogliosis in offspring brains, indicating the increased risk of age-related neurodegenerative diseases onset. Maternal inhalation exposure to CB NPs induced dose-dependent reactive astrogliosis in the cerebral cortex and hippocampus, and an increase in the glial fibrillary acidic protein (GFAP) protein level in astrocytes of the offspring brain [22]. After maternal exposure to CB NPs, C57BL/6 J mice were found to have sex- and region-specific pathological abnormalities in the CC and cortex with a significant increase in the GFAP protein level [84]. After the intravenous injection of MWCNTs functionalized with 1, 2-distearoyl-sn-glycero-3-phosphoethanolamine-N-[amino(polyethylene glycol)-2000] (PL-PEG-NH_2_), larger-sized MWCNTs moved across the blood–placenta barrier (BPB), restricted the development of fetuses, and induced brain deformity, whereas SWCNT and smaller-sized MWCNTs showed no or less fetotoxicity. Furthermore, p53^-/-^ fetuses showed obvious brain deformity, indicating that CNTs might have a genetic background-dependent toxic effect on the normal development of the embryo [70].

*Metal oxide NPs and Metal NPs.* Gestational exposure to metal oxide led to the accumulation of NPs in the fetal brain, causing neurotoxicity and neurobehaviors associated with enhanced depression-like behaviors, impaired learning and memory, and autism spectrum disorder in offspring [19,23,74,85,86,87,88,89]. For example, oral exposure to ZrO_2_ NPs during gestation caused their accumulation of fetal brain; hence, they were dangerous to fetal brain development, especially in early pregnancy [18]. Continuously intragastric feeding of TiO_2_ NPs (6.5 nm) to mice from prenatal day 7 to postnatal day (PND) 21 led to brain retardation, retarded axonal and dendritic outgrowth, and impaired cognitive ability; it was related to the excessive activation of the extracellular signal-regulated kinase/mitogen-activated protein kinase (ERK1/2/MAPK) signaling pathway [19]. The subcutaneous injection of TiO_2_ NPs into pregnant mice altered the expression levels of genes related to CNS development and function [23]. Exposures to ZnO NPs during gestation also induced neurobehavioral abnormalities, such as impaired learning and memory ability, depression-like behaviors, and adverse effects on reflexive motor behavior. The exposure of pregnant Sprague–Dawley rats to ZnO NPs (500 mg/kg) via gavage for 18 consecutive days significantly elevated the concentration of zinc in offspring brains. Abnormal neuron ultrastructure, histopathologic changes, and imbalanced antioxidant status were observed in offspring brains, impairing learning and memory ability in adulthood [85]. Recently, Hawkins et al. reported that maternal exposure to metal NPs induced autophagy-induced developmental neurotoxicity. In this study, the intravenous injection of cobalt and chromium (CoCr) NPs (0.12 mg/mouse) into pregnant mice on GD10 or GD13 resulted in neurodevelopmental abnormalities, such as reactive astrogliosis and aggravated DNA damage, in the fetal hippocampus [21].

NP-induced neurotoxicity depends on the physicochemical parameters of NPs, pregnancy stages, and administration procedures. The dissolution of NPs (i.e., Ag NPs or ZnO NPs) plays an important role in fetotoxicity after maternal exposure.

### 2.3. Reproductive Toxicity

Regarding the reproductive toxicity, reproductive dysfunction in male offspring after maternal exposure to NPs has been reported (Table 3).

*Ambient fine or ultrafine particles.* The intratracheal administration of Asian sand dust (ASD, 0.91–1.7 μm, 200 μg/mouse) on GD8 and GD15 into pregnant ICR mice induced significantly decreased testis and epididymis weight in offspring. Pathological alterations included the vacuolation of seminiferous tubules and cellular adhesion of seminiferous epithelia, which resulted in significantly decreased daily sperm production (DSP) in offspring [104]. Gestational exposure to DEPs disrupted testicular function and affected gonadal development in offspring, which included decreased seminal vesicle and prostate coefficients, loss of germ cells in seminiferous tubules, and increased DNA fragmentation rate of sperm [105,108].

*Carbon-based nanomaterials.* Most in vivo findings indicated that in utero exposure to CB NPs and MWCNTs might affect the reproductive system and function in male/female offspring [22,70,82,83,84]. CB NP-induced male reproductive disorders depended on different administration routes and doses. In detail, after pregnant mice were intratracheally instilled with CB NPs (200 μg/mouse, 14 nm) on GD8 and GD15, partially vacuolated seminiferous tubules and reduced cellular adhesion in seminiferous epithelia were observed 5, 10, and 15 weeks after birth, with DSP significantly decreased by 47%, 34%, and 32%, respectively [25]. Similar reduced DSP appeared in the second generation (F2) after prenatal exposure to CB NPs (200 μg/mouse) on GD7, 10, 15, and 18 [28]. However, maternal inhalation of CB NPs at occupationally relevant concentrations did not change male reproductive function even in four generations of offspring mice [24]. Treatment with MWCNTs and graphene quantum dots also did not induce alterations in the DSP and male reproductive ability of the exposed male offspring [27,71].

*Metal oxide NPs.* Previous investigations indicated that in utero exposure to ZnO NPs, TiO_2_ NPs, Fe_2_O_3_ NPs, and SiO_2_ NPs affected male reproductive ability and the reproductive function of offspring [28,59,60,106,107]. The subcutaneous injection of TiO_2_ NPs into pregnant mice resulted in alterations in spermatogenesis and impairment in the testis of offspring. TiO_2_ NPs were transferred to the offspring and accumulated in Sertoli cells, Leydig cells, or spermatids of offspring testes, leading to reduced DSP and the number of Sertoli cells, disorganized and disrupted seminiferous tubules, and failed spermiation from Sertoli cells and sperm motility [106]. In contrast, inhalation exposure to TiO_2_ NPs during gestation at the total doses of 0.5, 5, 50, and 500 μg did not affect the male reproductive function in two generations of male offspring mice [28]. The oral administration of ZnO NPs (50 and 100 mg/kg) and SiO_2_ NPs (250 mg/kg) triggered alterations in spermatogenesis and pathological changes in testis, such as epithelial vacuolization, cellular adhesion of epithelia, and decreased seminiferous tubule diameter [59]. Similarly, Di Bona et al. reported a charge- and dose-dependent effect on the reproduction of offspring by prenatal exposure to Fe_2_O_3_ NPs. Polyethyleneimine Fe_2_O_3_ NPs^(+)^ treatment induced morphological alterations in offspring uteri and testis during organogenesis, while poly(acrylic acid) Fe_2_O_3_ NPs^(−)^ treatment only induced mild alterations in the offspring uterine cavity [60]. The aforementioned data demonstrated that not only the accumulation of NPs in reproductive tissues but also the reproductive toxicity induced in offspring depended on administration routes, doses, exposure windows, and particle charge.

### 2.4. Immunotoxicity

The innate immune system of an organism provides the first line of defense against particulate materials. Pregnant women and developing fetuses are populations susceptible to immunotoxicity induced by xenobiotic pollutants. Developmental immunotoxicity induced by various NPs in mice is summarized in Table 4. 

*Ambient fine or ultrafine particles.* In recent years, detrimental effects on the immune development of the offspring after maternal exposures to PM have received more attention [29,30,109]. After maternal exposure to ultrafine particles derived from combustion (containing persistent free radicals) via oropharyngeal aspiration on GD 10 and 17, the development of pulmonary T helper and T regulatory cells was suppressed in the infant offspring at the age of 6 days. At the age of 6 weeks, the percentage of Th2 and T regulatory cells recovered to the control levels, while the percentage of Th1 and Th17 cells decreased. Maternal exposure to combustion-derived ultrafine particles aggravated postnatal asthma severity, in association with the suppression of pulmonary Th1 development and systemic oxidative stress in exposed mothers [29]. The intranasal administration of community-sampled PM into pregnant C57BL/6 mice on GD14, GD16, and GD18 altered the immune cell development of offspring mice in a sex-specific manner [30]. Furthermore, pregnant BALB/c mice intranasally instilled with DE on GD14 caused increased asthma susceptibility or transgenerationally transmitted asthma susceptibility in offspring [110].

*Carbon-based Nanomaterials and Metal NPs.* El-Sayed et al. reported the adverse effects on the immune system of offspring after gestational exposure to CB NPs [32,111]. After pregnant ICR mice were instilled intranasally with CB NPs (14 nm, 95 μg/kg) on GD9 and GD15, CB NPs triggered alterations in thymocyte and splenocyte phenotypes, and upregulated *Traf6* gene expression in the thymus. Therefore, respiratory exposure to CB NPs in middle and late gestation might cause allergic or inflammatory effects in male offspring mice [32]. When exposure occurred during early gestation, CB NPs suppressed the development of the offspring immune system partially, as characterized by decreased numbers of CD3^+^, CD4^+^, and CD8^+^ cells, and upregulated *Il15* (in male offspring), *Ccr7* and *Ccl19* (in female offspring) gene expression [111]. In one study investigating the developmental immunotoxicity induced by metal NPs in offspring, pregnant C57BL/6 mice were exposed to Cu NPs (15.7 nm, 3.5 mg/m^3^) via inhalation from GD3–GD19 [31]. Among 84 genes involved in T cell immune responses in spleens of pups, 14 genes were significantly upregulated and 11 genes were significantly downregulated at the age of 7 weeks, indicating that prenatal inhalation exposure induced significant immunomodulatory effects in offspring. Despite no translocation of NPs into the fetus, Cu NPs induced maternal pulmonary inflammation, subsequently leading to the disruption of the Th1/Th2 balance in the developing fetus.

### 2.5. Respiratory Toxicity

Epidemiological studies have proved that maternal exposure to ambient PMs during the gestational period is associated with disturbed lung development and increased risk of respiratory diseases after birth. So far, investigations have mostly focused on PMs; only a few have explored metal oxide NPs and metal NPs.

*Ambient fine or ultrafine particles.* The exposure of pregnant C57/BL6 mice to combustion-derived PMs via oropharyngeal aspiration aggravated asthma development in offspring mice [29]. This aggravation of asthma in offspring might be due to systemic oxidative damage in exposed mothers and Th1 maturation in offspring lungs. In utero exposure to PMs from residential roof spaces induced impairment in somatic growth, leading to reduced lung volume and dysfunction of male offspring lungs [30]. Yue et al. reported that increased inflammation in the respiratory system of the fetus and offspring was observed after maternal exposure to PM_2.5_ (3 mg/kg) by oropharyngeal aspiration every other day from GD1–GD17 [34].

*Metal NPs and Metal oxide NPs.* Maternal exposure to metallic NPs during pregnancy, such as TiO_2_ NPs, CeO_2_ NPs, and Ag NPs, might lead to the transfer of NPs from the mother to the fetal lung and induce impairment in lung development and even a lasting effect in adulthood [35,113]. For example, NPs translocated across the placental barrier and deposited in the offspring lungs after the mice were orally exposed to TiO_2_ NPs (21 nm) daily for 7 days during pregnancy [113]. The exposure induced delayed saccular development and inflammatory lesions in offspring lungs, accompanied by deficient septation, thickened mesenchyme between the saccules, atypical lamellar inclusions, macrophage infiltration, and thickened primary septa. In another study, the exposure of TiO_2_ NPs, CeO_2_ NPs, and Ag NPs resulted in irreversible impairment in offspring lungs regardless of the chemical properties of NPs with alterations in lung development-related genes and proteins in the fetal and alveolarization stages [35].

In summary, the aforementioned results provided a comprehensive understanding on how the key parameters of particles affected NP-induced fetotoxicity, including neurotoxicity, reproductive toxicity, developmental immunotoxicity, and respiratory toxicity, in offspring (Figure 2). However, the evaluation of fetotoxicity in other organs such as the liver, and cardiac function, feto-placental development, microvascular function, and fetal metabolism need further investigation.

## 3. Transplacental Transfer of NPs

The placenta is a vital organ connecting the mother and the fetus in mammals. During gestation, the placenta not only is the site for the exchange of substances, metabolism, secretion, and excretion, but also protects the embryo/fetus from the harmful environmental pollutants. Although anatomical differences in the placenta exist among species, the most commonly selected models used to explore transplacental transfer ability and fetotoxicity are human and murine placenta models. Different from human term placenta, the mature murine placenta consists of three trophoblast layers, including the decidua, junctional zone, and labyrinth, constituting the primary barrier [114,115,116]. Any structural and functional abnormalities in the placenta may impair its barrier function and affect fetal development. Therefore, an in-depth understanding of placental transfer and transplacental transport mechanisms is of great significance. The following section briefly discussed the maternal–fetal transfer of NPs and the underlying transport mechanisms.

### 3.1. Maternal–Fetal NP Transfer

Direct particle translocation across the placental barrier has been considered as the most potential mechanisms underlying NP-induced fetotoxicity [117]. Previous studies demonstrated that CB NPs, CNTs, PS NPs, Si O_2_ NPs [118], TiO_2_ NPs [118], ZnO NPs [11,48], ZrO_2_ NPs [18,119], CeO_2_ NPs, Au NPs [120,121,122], Ag NPs, and quantum dots (QDs) [15,123] could cross the placental barrier and induce fetotoxicity. Properties of NPs, maternal conditions, exposure schedule, and modification of NPs by biofluid components affected the placental transfer efficiency of NPs [124,125].

Particle size is a pivotal determinant regulating NP translocation across the placental barrier [15,16,48,121,126]. The maternal–fetal transfer of some spherical NPs, including Au NPs [120,121,122], SiO_2_ NPs [16,127], ZnO NPs [48], and QDs [15,123], was affected by the size. Surface modification also affects the transfer efficiency of NPs. For example, carboxylate-modified Au NPs showed higher uptake and deeper penetration than PEG-modified Au NPs [122]. SiO_2_ NPs coated with -COOH/-OH or QDs coated with -PEG/-SiO_2_ shell prevented or reduced NP translocation into the fetus [15,123]. Additionally, surface charge and material type played an important role in the transplacental translocation of NPs [63,128,129,130]. Maternal physiological and pathological conditions during exposure could influence the transfer of NPs. Maternal intrauterine inflammation significantly increased the maternal–fetal transfer of Au NPs (3 nm and 13 nm) [120]. Finally, secondary modifications of NPs by biofluid components, such as the formation of protein corona, affected the placental transfer of NPs [124,131].

### 3.2. Transplacental Transport Mechanisms

So far, investigations regarding the transfer mechanisms of NPs across the placental barrier have been limited [18,121,128]. The possible transport pathways include passive diffusion, active transport, and endocytic pathways (Figure 3) [132,133,134]. Passive diffusion across the placenta is the predominant transport mechanism for small NPs [133]. In the in vitro BeWo b30 transfer model, the placental translocation of charged polystyrene (PS) NPs (50 nm) mainly depended on passive diffusion [129]. In contrast, in the ex vivo human placental perfusion model, similar PS NPs (50–300 nm) showed an increased transfer of NPs from fetal to maternal direction through active, energy-dependent transplacental transport mechanisms [128]. Endocytosis has been proposed as another transport mechanism underlying the maternal–fetal transfer of NPs through the placenta [126,132]. Maternal exposure to Au NPs (50 nm) or ZrO_2_ (16 nm)-induced maternal–fetal transfer of NPs across the placenta via the upregulation of clathrin- and caveolin-mediated endocytosis [18,135]. In an early study, it was hypothesized that diffusion, transtrophoblastic channels, and/or receptor-mediated endocytosis were the potential mechanisms involved in the translocation of AuNPs (1.4- and 18-nm) [121]. In addition, the physicochemical characteristics of NPs, including chemical composition, hydrophilicity, and surface charge, might influence the transport of NPs to the fetus through interacting with the receptors of trophoblast cells [38].

## 4. Molecular Mechanisms Involved in NP-Induced Fetotoxicity

Maternal NP exposure may induce fetotoxicity by the direct translocation of NPs to the fetus or indirect NP-induced maternal and placental mediators [37,38,117]. During the critical window of fetal development, the factors interfering with the whole process of gestation (maternal condition, placental function, and developing fetus), contribute to detrimental fetal outcomes. This section discussed different molecular mechanisms involved in NP-induced developmental toxicity.

### 4.1. Oxidative Stress and Inflammatory Responses

Reactive oxygen species (ROS) generation and inflammatory responses are generally considered as the main mechanisms in NP-induced fetotoxicity [136]. First, this may occur when NPs enter into maternal or fetal tissues. For example, prenatal exposure to CB NPs, and TiO_2_ NPs during pregnancy triggered oxidative damage to nucleic acids and lipids, and induced reactive astrocytes via the overexpression of GFAP and Aquaporin 4 (Aqp4) in offspring brain [22,84]. Inhaled NPs localized in the lungs could generate ROS and inflammatory responses, which became a potent maternal modulator for fetal development. Prenatal exposure to TiO_2_ NPs (21 nm, 40 mg/m^3^) 1h/day from GD8–GD18 induced the deposition of NPs in lungs and long-term maternal lung inflammation, resulting in a sex-specific neurobehavioral alteration in offspring [74]. Second, the balance between oxidants and antioxidants is vital for maintaining physiological homeostasis in the placenta. Once NPs are taken up by the placental cells, the induction of excessive ROS results in oxidative stress and placental inflammation, which has been hypothesized to represent one indirect mechanistic pathway of NP-induced placental dysfunction and fetotoxicity. The intravenous injection of SiO_2_ NPs (70 nm, 0.04 mg/g) into pregnant mice from GD13–GD14 upregulated nucleotide-binding oligomerization domain-like receptor (NLR) family pyrin domain-containing 3 (NLRP3) inflammasome and significantly increased the levels of placental inflammatory cytokines (IL-1β, IL-6, TNF-α, and CCL2) [64]. This exposure during gestation triggered NLRP3-mediated placental inflammation and placental dysfunction due to ROS generation and oxidative stress, resulting in pregnancy-related complications. Increased ROS levels in malformed fetuses and their placentas were observed following the retrobulbar injection of uo-SWCNTs from GD6 [17]. After pregnant mice were intravenously injected with o-SWCNTs (20 mg/kg), increased levels of ROS and decreased levels of vascular endothelial growth factor (VEGF) led to narrowed vessels and decreased blood vessels in the placenta, which induced miscarriage and fetal growth retardation [65]. Maternal exposure to other NPs such as SiO_2_ NPs, SWCNTs, and TiO_2_ NPs, induced oxidative stress and a high incidence of detrimental developmental outcomes [17,64,89]. These results proved the involvement of oxidative stress and inflammation in NP-induced fetotoxicity.

### 4.2. DNA Damage

DNA damage is generally thought to be responsible for initiating fetal developmental abnormalities caused by environmental chemicals. Maternal exposure to NPs during pregnancy may cause severe DNA damage, such as DNA strand breaks, DNA deletions, mutations and oxidative DNA adducts. SWCNT-50 can induce p53-dependent fetotoxicity and brain deformity in p53^-/-^ fetuses, which is related to MWCNT 50-induced DNA damage [70]. Maternal inhalation exposure to CB NPs (42 mg/kg, 1h/day) from GD8–GD18 caused DNA strand breaks in the exposed offspring, while prenatal TiO_2_ NPs exposure altered the gene expression of the retinoic acid signaling pathway in female rather than male mice [43,137]. Transplacental Au NP exposure (100 nm, 3.3 mg/kg) during organogenesis indicated size-dependent clastogenic and epigenetic alterations in fetuses, while transplacental DEP exposure caused DNA deletions [69,138]. Maternal NP exposure to CoCr NPs aggravated DNA damage in the fetal hippocampus through autophagy dysfunction and the release of IL-6 [21].

### 4.3. Apotosis

Maternal exposure to SiO_2_ NPs, TiO_2_ NPs, Au NPs, QDs, functionalized CNTs, and carboxylate-modified polystyrene beads during pregnancy induced cell apoptosis in spongiotrophoblasts of the placenta [15,16,120,139,140], the fetal hippocampus [19,23,92,93], and fetal liver [70,141,142], which led to placental dysfunction and poor developmental outcomes.

For example, gestational exposure to TiO_2_ NPs (<25 nm) via gavage from GD1–GD13 triggered apoptosis, dysregulated vascularization, and proliferation, which significantly impaired placental growth and development [140]. In addition, TiO_2_ NPs suppressed axonal or dendritic outgrowth, through the induction of apoptosis in the offspring hippocampus [19,23,92,93,107]. Similarly, apoptosis was also observed in the 6-week offspring brains following maternal exposure to ultrafine CB [143]. In another study, functionalized MWCNTs directly triggered p53-dependent apoptosis and cell cycle arrest through DNA damage, suggesting genetic background-dependent developmental abnormalities [70]. Carboxylate-modified polystyrene beads can pass through the placenta into fetal organs, and induced trophoblast apoptosis [139]. Other nanomaterials, including SiO_2_ NPs, ZrO_2_ NPs, and Au NPs likewise induced pathological and structural alterations in apoptosis and apoptosis-related factors in the placenta following gestational NP exposure [16,18,65,120].

### 4.4. Autophagy

NP-induced autophagy has been widely reported [144,145]. For example, both MWCNTs and two-dimensional MoS_2_ nanosheets induced autophagy relying on the variation in surface chemistry and shape [145,146,147]. Throughout pregnancy, autophagy is involved in oocytogenesis [148], implantation [149], placentation [150,151], embryogenesis [152,153], preeclampsia [150,154] and preterm delivery [155]. Especially during placental development, autophagy, mediated by alterations in oxygen and glucose levels in cytotrophoblast cells, participates in placentation and placental-related diseases, ultimately affecting fetal development. Some studies demonstrated that autophagy might alleviate the toxicity of platinum NPs in trophoblasts [156] and protect the fetal brain under short-term food deprivation [157]. However, autophagic dysfunction in the placenta is associated with fetal diseases such as fetal growth retardation and neonatal encephalopathy [158]. CoCr NPs induced autophagic dysfunction in BeWo cell barriers, contributing to the altered differentiation of human neural progenitor cells and DNA damage in the derived neurons and astrocytes [21]. TiO_2_ NPs induced autophagy and possible placental dysfunction in HTR-8/SVneo cells through the activation of endoplasmic reticulum stress [159,160]. MiR-96-5p and miR-101-3p might act as potential targets to reverse autophagy and placenta dysfunction induced by TiO_2_ NPs in a human trophoblast model [160]. In addition, abnormal autophagy disrupted organism development and differentiation during embryogenesis. Zhou et al. demonstrated that maternal exposure to TiO_2_ NPs induced excessive autophagy with significant enhancement in expression levels of autophagy-related factors and LC3 II/LC3I, resulting in the severe suppression of dendritic outgrowth in offspring hippocampal neurons [93].

The aforementioned studies revealed that NPs might disrupt placental development and embryo–fetal development via autophagy. The placenta acted as a vital support unit for the developing fetus and played a pivotal role in regulating fetal development such as fetal brain [161]. Therefore, placental dysfunction mediated by autophagy and its molecular events might be one of the mechanisms underlying NP-induced fetal developmental abnormalities. Considering its two-way regulation and limited data regarding autophagy in the placenta, further studies should be conducted to explore the role of autophagy in NP-induced fetotoxicity.

In addition, NPs may induce fetotoxicity via interfering with endocrine signaling [162,163], vascular signaling [16,56,164], placental Toll-like receptors [165], and other cellular signaling pathways. The major underlying molecular mechanisms involved in NP-induced fetotoxicity are summarized in Figure 4.

## 5. Conclusions and Perspectives

The evaluation of NP-induced toxicity across placental barriers has become increasingly important with the increased application of nanomaterials in medical products. Maternal exposure to NPs during pregnancy led to adverse gestational parameters, neurotoxicity, reproductive toxicity, immunotoxicity, and respiratory toxicity in offspring. The characteristics and functionalization of NPs, as well as maternal conditions and exposure routes, play crucial roles in NP-induced fetotoxicity. Oxidative stress and inflammation, DNA damage, apoptosis, and autophagy have been considered as the main mechanisms underlying the NP-induced fetotoxicity. However, the detailed mechanisms underlying the embryo–fetal toxicity need further investigation due to the complexity in the pharmacokinetics of NPs and interactions with physiological systems, in particular during pregnancy.

The nano-safety evaluation during pregnancy has been extensively explored. However, more systematic investigations should be conducted on the following issues in the future. First, the NP-induced fetotoxicity focused on the CNS, reproductive, respiratory and immune systems. More studies on other systems, including cardiac function, early placental development, microvascular function, and fetal metabolism should be performed. Second, NP libraries and more effective placental transfer models (in vitro/ex vivo perfusion models and animal models) should be constructed to systematically elucidate how NP translocation is dependent on the physiochemical properties of NPs and clarify the underlying mechanisms. Third, the exposure route adopted in current investigations, especially intravenous administration, is not the representative route for maternal environmental and occupational exposure. Hence, strategies for exposure assessment in the environment and at the workplace should be elaborated, which is the base point for the establishment of occupational exposure limits for NPs.

## Figures and Tables

**Figure 1 nanomaterials-11-00791-f001:**
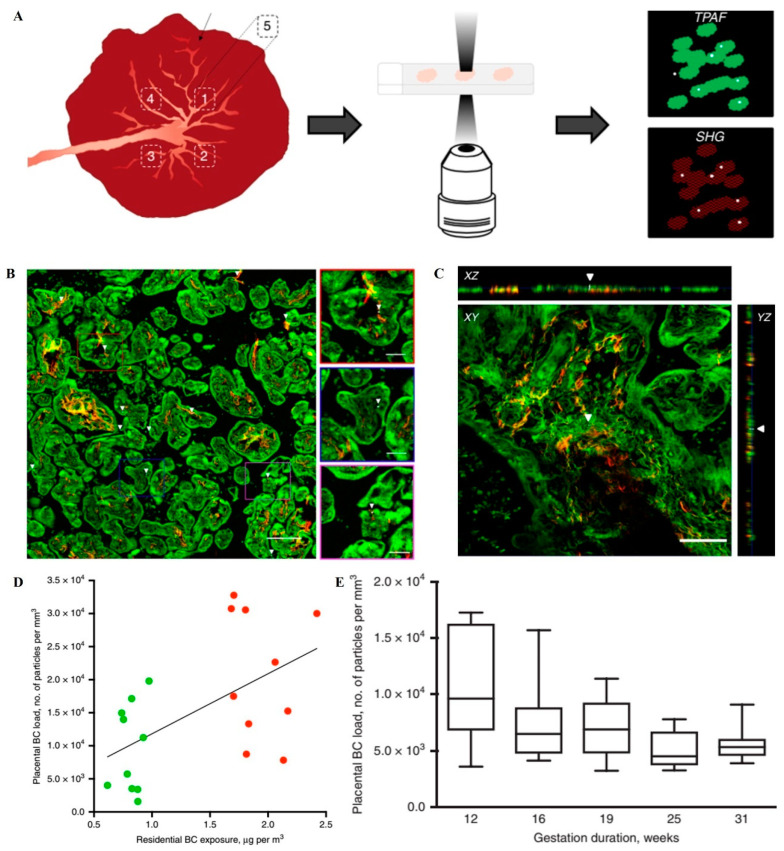
The protocol for black carbon detection in the placenta and evidence of them at the fetal side of human placenta. (**A**) The detection of BC particles in placenta. The white light produced by the BC naturally present in the tissue (white dots) is detected along with the simultaneous generation and detection of two-photon excited autofluorescence (TPAF) of the cells (green) and second harmonic generation (SHG) from collagen (red). (**B**) The evidence of ambient BC particles at the fetal side of human placenta. White light (WL) generation originating from the BC particles (white and further indicated using white arrowheads) under femtosecond pulsed laser illumination (excitation 810 nm, 80 MHz, 10 mW laser power on the sample) is observed. Scale bar: 30 μm. (**C**) Validation of the carbonaceous nature of the identified particles inside the placenta. XY-images acquired throughout a placental section in the z-direction and corresponding orthogonal XZ-projections and YZ-projections showing a BC particle (white and indicated by white arrowheads) inside the tissue (red and green). Scale bar: 50 μm. (**D**) Correlation between placental load and residential exposure of CB particles throughout the whole pregnancy. The line is the regression line. Green and red dots indicate low (n = 10 mothers) and high (n = 10 mothers) exposed mothers. (**E**) BC load in placentas from spontaneous preterm births (n = 5). The whiskers indicate the minimum and maximum value and the box of the box plot illustrates the upper and lower quartile. The median of spreading is marked by a horizontal line within the box. Reproduced with permission [36]. Copyright Springer *Nature*, 2019.

**Figure 2 nanomaterials-11-00791-f002:**
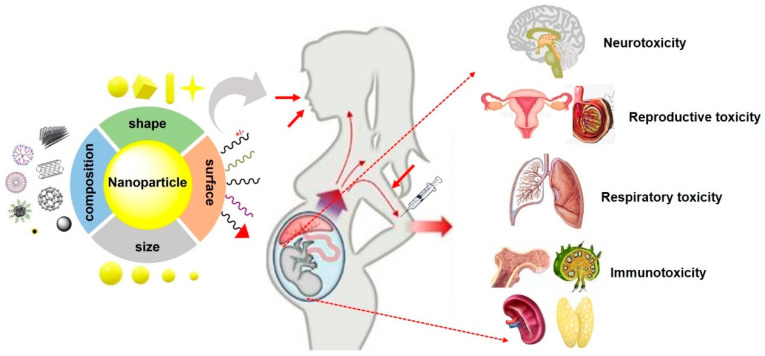
Typical fetotoxicity potentially induced by various NPs. The properties of NPs such as size, shape, composition and surface chemistry are key parameters that affect fetotoxicity following maternal exposure NPs during pregnancy. In addition, maternal conditions and exposure routes, also play crucial roles in NP-induced fetotoxicity.

**Figure 3 nanomaterials-11-00791-f003:**
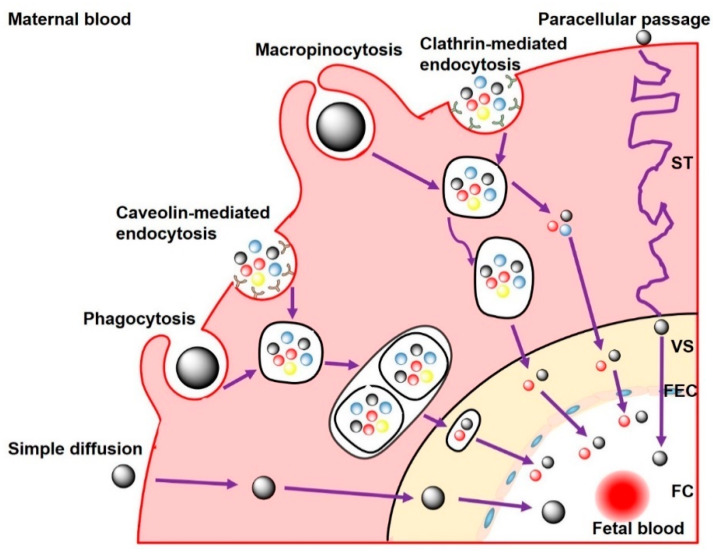
Scheme showing the transplacental transport of NPs. NPs cross the blood–placental barrier by paracellular passage and transcellular passage. Very small NPs can cross syncytiotrophoblasts (STs) through placental channels, and enter villous stroma (VS). Diffusion may then occur through the fetal endothelial cells (FECs) into the lumen of the fetal capillaries (FCs). NPs may also be taken up by STs via phagocytosis, clathrin-/caveolae-mediated endocytosis, and macropinocytosis, and eventually enter the fetal blood.

**Figure 4 nanomaterials-11-00791-f004:**
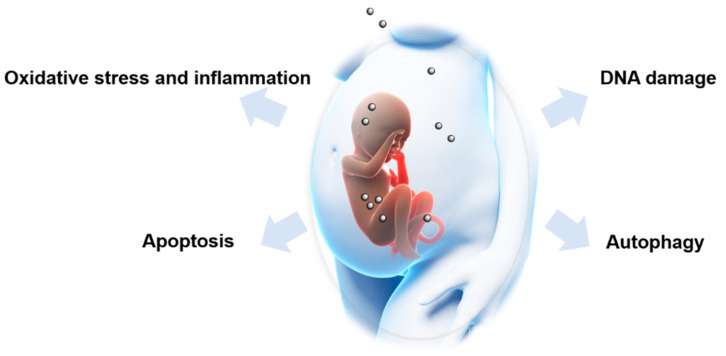
Possible mechanisms involved in nanoparticle-induced fetotoxicity. Oxidative stress and inflammation, DNA damage, apoptosis and autophagy are major mechanisms underlying nanoparticle-induced fetotoxicity.

**Table 1 nanomaterials-11-00791-t001:** Summary of gestational parameters induced following maternal exposure to nanoparticles (NPs).

NPs	Animals/Exposure Route	Fetotoxicities	Ref.
CeO_2_ NPs 3–5 nm	Mouse, i.v., 5 mg/kg, GD5, 6, and 7	Decreased number and pups’ weight; increased fetal resorption rate	[10]
ZnO NPs 13 nm	Mouse, i.g., GD7-GD16	Intrauterine growth retardation	[11]
Ultrafine particles	Mouse, intratracheal instillation, 400 μg/kg, GD7, 9, 11, 15 and 17	Embryo reabsorption, decreased fetal weight and altered blood pressure in the offspring	[12]
QDs	Rat, inhalation exposure, 5,1 nmol/rat, GD5–GD19	Growth restriction in offspring	[13]
Ag NPs 18–20 nm	Mouse, intranasal instillation, 640 μg/m^3^, GD1–GD15	Increased number of resorbed fetuses	[14]
CdSe/ZnS QDs, 20 nm CdSe QDs, 15 nm	Mouse, i.v., 0.1 nmol/mouse, GD17 and 18	Fetus malformation, hampered growth	[15]
SiO_2_ NPs, 70 nm TiO_2_ NPs, 35 nm	Mouse, i.v., SiO2 NPs: 0.2, 0.4, 0.8 mg/mouse, TiO_2_ NPs: 0.8 mg/mouse, GD16 and 17	Smaller fetuses	[16]
p,o,uo-SWCNTs	Mouse, the retrobulbar injection, 0.01~30 μg/mouse, GD6	Retarded limbs and snout development	[17]
Diesel engine exhaust (DEP), 69 nm	Rabbit, nose-only inhalation, 2 h/day, 5 days/week, GD3–GD27	Growth retardation at mid gestation with decreased head length and umbilical pulse	[41]
NP-enriched DE, 22–27 nm	Rat, intranasal instillation, 148.86 μg/m^3^, 3.10 μg/m^3^, 5 h/day, GD1–GD19	Increased fetal weight and decreased crown-rump length	[42]
CB NPs 14 nm	Mouse, intratracheal instillation, 11, 54, 268 μg/mouse, GD7, 10, 15 and 18	Induced more DNA strand breaks in the liver of their offspring	[43]
MWCNTs	Mouse, i.p. or intratracheal instillation, 2, 3, 4, 5 mg/kg, GD9	Increased the number of external malformation and skeletal malformation in fetuses	[44]
SWCNTs	Rat, intratracheal instillation or i.v., 100 μg/kg, GD17–GD19	Induced vasoconstriction and reduced fetal growth	[45]
rGO	Mouse, i.v., 6.25, 12.5, 25 mg/kg, GD6 or 20	Caused malformation in fetuses	[46]
ZnO NPs 30 nm	Mouse, i.g., 20, 60, 180, 540 mg/kg, GD11–GD18	Fetal growth retardation, decreased fetal number	[47]
ZnO 13, 57 or 1900 nm	Mouse, i.g., GD7–GD16	Decreased birth weight	[48]
ZnO NPs 44.2 nm	Rat, i.v., 5, 10, 20 mg/kg, GD7–GD21	Increased the number of dead fetuses and decreased fetal weight	[49]
ZnO NPs 20 nm	Mouse, i.g., 100, 200, 400 mg/kg, GD5–GD19	Significant decrease in fetal weight for 400 mg/kg exposure group	[50]
ZnO NPs 35 nm	Rat, i.g., 500 mg/kg, 2 weeks before mating to PND 4	Reduced fetal weight and increased fetal resorption of pups	[51]
TiO_2_ NPs 21 nm	Rat, inhalation exposure, 12 mg/m^3^, 6 h/exposure, 6 days, GD11–GD16	Significantly decreased pup’s weight and placental efficiency	[52]
TiO_2_ NPs 6.5 nm	Mouse, i.g., 25, 50, 100 mg/kg, GD1–GD18	Inhibited the crown-rump length, fetal weight, the number of live fetuses and fetal skeleton development	[53]
SiO_2_ NPs 25, 60, 115 nm	Mouse, i.v., 3, 30, 200 μg/mouse, GD6, 13 and 17	Decreased the resorbed number at the dose of 200 μg/mouse for 25 nm NPs	[54]
CdO NPs 11–15 nm	Mouse, inhalation exposure, 100 μg/m^3^, every other day or 230 μg/m^3^ daily, 2.5 h/day, GD5–GD17	Decreased incidence of pregnancy, fetal length, and neonatal growth	[55]
Ag NPs 20, 110 nm	Rat, i.v., 200 μg/rat, GD17–GD19	Fetal growth restriction	[56]
Ag NP 10 nm	Mouse, i.v., 66 μg/mouse, GD8–GD10	Embryonic growth restriction	[57]
Au NPs 2, 15, 50 nm	Mouse, i.v., 2 mg/kg, or 0.5–10 mg/kg, GD4–GD6	Disturb embryonic development in a size- and concentration-dependent manner.	[58]
ZnO NPs SiO2 NPs	Mouse, i.g., ZnO NPs: 50, 100, 300 mg/kg, SiO2 NPs: 50, 100 mg/kg, GD5–GD19	Miscarriages and adversely affected the developing fetus	[59]
PEI-Fe_2_O_3_-NPs, 28 nm PAA-Fe_2_O_3_-NPs, 30 nm	Mouse, i.p., 10, 100 mg/kg, GD9, 10 and 11	High dose exposure led to charge-dependent fetal loss, morphological alterations in uteri	[60]
TiO_2_ NPs 20 nm	Rat, inhalation exposure, 10 mg/m^3^, 6 h/exposure, 6 days, GD5–GD19	Altered fetal epigenome	[61]
QDs 1.67, 2.59 or 3.21 nm	Mouse, i.p., 5, 10, 20 mg/kg, GD14	Decreased survival rate, body length, body mass and disturbed ossification of limbs	[62]
Fe_2_O_3_ NPs 28–30 nm	Mouse, i.p., 10 mg/kg, GD10–GD17	Increased fetal death	[63]
SiO_2_ NPs 70 nm	Mouse, i.v., 25, 40 mg/kg, GD13–GD14	Pregnancy complications	[64]
o-MWCNTs	Mouse, i.v., 20 mg/kg, GD4, 11 and 15	Induced maternal body weight gain and abortion rates dependent on pregnancy times	[65]
CNTs	Mouse, i.v., 10 μg/mouse, GD6 and 15	Occasional teratogenic effects	[66]
fCNTs	Mouse, i.g., 10 mg/kg, GD9	Increased the number of resorbed fetuses; fetal morphological and skeletal abnormalities	[67]
TiO_2_ NPs Ag NPs	Mouse, i.g., 10, 100, 1000 mg/kg, GD9	Increase fetal mortality	[68]
DEPs	Mouse, i.g., 31.25, 62.5, 125, 250, 500 mg/kg, GD11–GD16	Increased the frequency of DNA deletions in fetus and offspring	[69]

Abbreviations: GD, gestational day; PND, postnatal day; i.v., intravenous; i.g., intragastrical; i.p., intraperitoneal; Ref., references; QDs, quantum dots; SWCNTs, single-walled carbon nanotubes; DE, diesel exhaust CB, carbon black; MWCNTs, multi-walled carbon nanotubes; CNTs, carbon nanotubes; rGO, reduced graphene oxide; PEI, polyethyleneimine; PAA, poly(acrylic acid).

**Table 2 nanomaterials-11-00791-t002:** NP-induced neurotoxicity following maternal exposure during pregnancy.

NPs	Animals/Exposure Route	Neurotoxicity	Ref.
TiO_2_ NPs 6.5 nm	Mouse, i.g., 1.25, 2.5, 5 mg/kg, GD7–PND21	Retarded axonal and dendritic outgrowth	[19]
DE	Mouse, s.c., 0.5, 1 mg/mL, GD5, 8, 11, 14 and 17	Increased glial-fibrillary acidic protein level in the corpus callosum and cortex	[20]
CoCr NPs	Mouse, i.v., 0.12 mg/mouse, GD10 and 13	Neurodevelopmental abnormalities with reactive astrogliosis and increased DNA damage in fetal hippocamus	[21]
CB NPs	Mouse, intranasal instillation, 2.9, 15, 73 μg/kg, GD5 and 9	Reactive astrogliosis	[22]
TiO_2_ NPs 25–70 nm	Mouse, s.c., 1 μg/μL, 100 μL, GD7, 10, 13 and 16	Changed gene expression related to neurotransmitters and psychiatric diseases in newborns	[23]
SWCNTs	Mouse, i.v., 2 mg/kg, GD11, 13, and 16	Obvious brain deformity	[70]
TiO_2_ NPs 21 nm	Mouse, inhalation exposure, 42 mg/m^3^, 1 h/day, GD8–GD18	Moderate neurobehavioral alterations in offspring mice	[74]
CB NPs 100–300 nm	Mouse, airway instillation, 0, 4.6, 37 mg/m^3^, 15 days, GD4–GD18	Denaturation of perivascular macrophages and reactive astrocytes	[82]
CB NPs 84 nm	Mouse, intranasal instillation, 190 μg/kg, GD5 and 7	Gene dysfunction in the frontal cortex in offspring mice	[83]
CB NPs 84.2 nm	Mouse, intranasal instillation, 95 μg/kg, GD5 and 9	Astrogliosis in the offspring brain	[84]
ZnO NPs 30 nm	Rat, ig., 500 mg/kg, GD2–GD19	Learning and memory impairment in the offspring brain	[85]
TiO_2_ NPs	Mouse, i.v., 100, 1000 μg, every second day, GD9	Autism spectrum disorder-related behavioral deficits in the offspring	[86]
TiO_2_ NPs 10 nm	Rat, i.g., 100 mg/kg, GD2–GD21	Impaired memory and decreased hippocampal cell proliferation in rat offspring	[88]
TiO_2_ NPs 5 nm	Rat, s.c., 1 μg/μL, 500 μL, GD7, 10, 13, 16 and 19	Oxidative damage in the brain of newborn pups, and the depressive-like behaviors during adulthood	[89]
Ag NPs 10 nm	Mouse, s.c., 0.2, 2 mg/kg, once every three days, GD1–GD21	Gender-specific depression-like behaviors in offspring	[90]
Ultrafine particles	Mouse, airway instillation, 92.69 μg/m^3^, 6 h/day, GD1–GD17	Neurodevelopmental disorders in offspring	[91]
TiO_2_ NPs 6.5 nm	Rat, i.g., 100 mg/kg, GD2–GD21, PND2–PND21	Impacted hippocampal neurogenesis and apoptosis in the offspring	[92]
TiO_2_ NPs 6.5 nm	Mouse, i.g., 1, 2, 3 mg/kg, GD1–PND21	Inhibited dendritic outgrowth of hippocampal neurons in the offspring mice	[93]
ZnO NPs 20–40 nm	Mouse, s.c., 4 h/day, PND4–PND7, PND10–PND13	Decreased ambulation score, hindlimb suspension score and degree of grip strength; increased degree of hindlimb foot angle	[94]
ZnO NPs 20–40 nm	Mouse, s.c., 0.5, 1 mg/mL, GD5, 8, 11, 14 and 17	Depressive-like behaviors in offspring	[95]
TiO_2_ NPs 20 nm	Mouse, i.p., 2 mg/mL, GD11– GD16	Decreased size and weight of fetus, a disrupted anatomical structure of the fetal brain, bulkier and abnormal shape of fetal liver	[96]
TiO_2_ NPs 170.9 nm	Rat, airway instillation, 10.4 mg/m^3^, 5 h/day, 4 days/week, GD7–GD20	Induced psychological deficits in male adulthood rat	[97]
Ag NPs 10 nm	Mouse, s.c., 0.2, 2 mg/kg, once every three days, GD1–GD21	Neurobehavioral disorders in the offspring	[98]
USPIO NPs	Mouse, i.v., 6.25, 12.5, 25 mg/kg, GD6 or 20	Abnormal fetal neurodevelopment	[99]

Abbreviations: GD, gestational day; i.v., intravenous; i.g., intragastrical; i.p., intraperitoneal; s.c., subcutaneous; USPIO, ultrasmall superparamagnetic iron oxide; Ref., references.

**Table 3 nanomaterials-11-00791-t003:** NP-induced reproductive toxicity following maternal exposure during pregnancy.

NPs	Animals/Exposure Route	Fetotoxicities	Ref.
CB NPs 14 nm	Mouse, intranasal instillation, 200 μg/kg, GD8, GD15	Alteration in reproductive function of male offspring	[25]
CB NPs 14 nm;	Mouse, intratracheal instillation, 268 μg/mouse, GD7, 10, 15 and 18	Lowered sperm production	[28]
TiO_2_ NPs 21 nm	Mouse, intranasal instillation, 42 mg/m^3^, 1 h/day, 840 μg/mouse, GD8–GD18	Lowered sperm production	[28]
SiO_2_ NPs ZnO NPs	Mouse, i.g., ZnO NPs: 0,50,100,300 mg/kg, SiO_2_ NPs: 0,50,250 mg/kg, GD15–GD19	Prominent epithelial vacuolization, decreased seminiferous tubule diameter in testis	[59]
PEI-NPs 28 nm PAA-NPs 30 nm	Mouse, i.p., 10, 100 mg/kg, GD9, 10 and 11	charge-dependent fetal loss, morphological alterations in uteri and testes of offspring	[60]
Asian sand dust	Mouse, intratracheal instillation, 200 μg/mouse, GD8 and 15	Partial vacuolation of seminiferous tubules and low DSP in immature offspring	[104]
Nanoparticle-rich DE	Rat, inhalation exposure, 148.86 g/m^3^, 1.83 × 10^6^ particles/cm^3^, GD2–GD20	Endocrine disruption after birth and suppression in testicular function	[105]
TiO_2_ NPs 35 nm	Mouse, s.c., 0.5, 5, 50, 500 μg/mouse, GD5, 8, 11, 14 and 17	A dose-dependent increase in the number of agglomerates in the offspring testes	[106]
TiO_2_ NPs 25–70 nm	Mouse, s.c., 1 mg/mL, 100 μL, GD3, 7, 10 and 14	Decreased daily sperm production in offspring	[107]

Abbreviations: GD, gestational day; i.g., intragastrical; i.p., intraperitoneal; s.c., subcutaneous; Ref., references; DSP, daily sperm production.

**Table 4 nanomaterials-11-00791-t004:** NP-induced immunotoxicity following maternal exposure during pregnancy.

NPs	Animals/Exposure Route	Fetotoxicities	Ref.
PMs	Mouse, oropharyngeal aspiration, 3 mg/kg, GD10, and 17	Inhibition of the development of pulmonary T helper and T regulatory cells of the infant offspring	[29]
PMs	Mouse, intranasal instillation, 95 μg/kg, GD14, 16, and 18	Inhibition of splenic T cell maturation in male offspring and alteration in early life immune development in a sex specific manner	[30]
Cu NPs	Mouse, inhalation exposure, 3.5 mg/m^3^, 4 h/day, GD4–GD20	Altered expression of several Th1/Th2 or other immune response genes in pups’ spleens	[31]
CB NPs 14 nm	Mouse, intranasal instillation, 95 μg/kg, GD9 and 15	Allergic or inflammatory effects in male offspring	[32]
PM_2.5_	Moue, inhalation exposure, 8 h/day, 6 days/week, 16 weeks	Alteration in immune microenvironment	[109]
DEPs 14 nm	Mouse, intranasal instillation, 50 μg/mouse, GD14	Increased allergic susceptibility in offspring	[110]
CB NPs 14 nm	Mouse, intranasal instillation, 95 μg/kg, GD9 and 15	Suppressed development of immune system of the offspring mice	[111]
DEPs	Mouse, i.g., 31.25, 62.5, 125, 250, 500 mg/kg, GD11–GD16	Increase in the frequency of DNA deletions in the mouse fetus and such genetic alterations in the offspring	[69]
DEPs	Mouse, intranasal instillation, 50 μg/mouse, GD14–GD15	Triggered transgenerational transmission of asthma risk	[112]

Abbreviations: GD, gestational day; i.g., intragastrical; Ref., references; PM, particulate matter.

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
