# Peer review of "Fetotoxicity of Nanoparticles: Causes and Mechanisms"

_nanomaterials, 2021, doi:10.3390/nano11030791_

Round 1
Reviewer 1 Report
Reviewer Comments to Author(s)
This review entitled “Fetotoxicity of Nanoparticles: Causes and Mechanism” summarizes adverse effects of nanoparticles on fetotoxicity. I feel that this review is well-written and valuable for readers in the fields of material and medical scientists. Although transplacental transport pathways of NPs are described in Figure 3, one point I would like to know is how NPs are accumulated in fetal tissues. Actively transferred from mother bodies or not? Other minor comments were as follows.
Line 70: What is the difference between NP-rich NR-DE and F-DE? I cannot understand that NP-removed F-DE has similar effects on fetal weight and decreased fetal crown-rump length to NR-DE.
Figure 1: How to image CBs as white dots?
Lines 178, 251, 268: mistypo (space)
Line 440: ex vivo (italic)
Reviewer 2 Report
To Authors
1) Abstract. Could you please add a brief description of the aim of article in the abstract?
2) Introduction. Could you please improve the description of the aim in the Introduction?
3) Could you please ameliorate the legends of Figures and Tables?
4) Could you please improve the Conclusions reporting a description of the NP-induced fetotoxicity on the CNS, reproductive, respiratory and immune systems?
